# Mortality after emergency unit fluid bolus in febrile Ugandan children

Brian Rice[1,2]*, Jessica Hawkins[3], Serena Nakato[2,4], Nicholas Kamara[4], on behalf of Global Emergency Care Investigator Group[2¶]

1 Department of Emergency Medicine, Stanford University, Palo Alto, California, United States of America, 2 Global Emergency Care, Shrewsbury, Massachusetts, United States of America, 3 Department of Anesthesia, Massachusetts General Hospital, Harvard Medical School, Cambridge, Massachusetts, United States of America, 4 Karoli Lwanga Hospital, Rukungiri, Uganda

¶ Membership of the Global Emergency Care Investigator Group is provided in the Acknowledgments.
* brice@stanford.edu

## Abstract

### Objectives

Pediatric fluid resuscitation in sub-Saharan Africa has traditionally occurred in inpatients. The landmark Fluid Expansion as Supportive Therapy (FEAST) trial showed fluid boluses for febrile children in this inpatient setting increased mortality. As emergency care expands in sub-Saharan Africa, fluid resuscitation increasingly occurs in the emergency unit. The objective of this study was to determine the mortality impact of emergency unit fluid resuscitation on febrile pediatric patients in Uganda.

### Methods

This retrospective cohort study used data from 2012–2019 from a single emergency unit in rural Western Uganda to compare three-day mortality for febrile patients that did and did not receive fluids in the emergency unit. Propensity score matching was used to create matched cohorts. Crude and multivariable logistic regression analysis (using both complete case analysis and multiple imputation) were performed on matched and unmatched cohorts. Sensitivity analysis was done separately for patients meeting FEAST inclusion and exclusion criteria.

### Results

The analysis included 3087 febrile patients aged 2 months to 12 years with 1,526 patients receiving fluids and 1,561 not receiving fluids. The matched cohorts each had 1,180 patients. Overall mortality was 4.0%. No significant mortality benefit or harm was shown in the crude unmatched (Odds Ratio [95% Confidence Interval] = 0.88 [0.61–1.26] or crude matched (1.00 [0.66–1.50])) cohorts. Adjusted cohort analysis (including both complete case analysis and multiple imputation) and sensitivity analysis of patients meeting FEAST inclusion and exclusion criteria all also failed to show benefit or harm. Post-hoc power calculations showed the study was powered to detect the absolute harm seen in FEAST but not the relative risk increase.

**Data Availability Statement:** Data cannot be shared publicly because of the lack of a formal data use and sharing policy for public access from the Mbarara University of Science and Technology.

Data are available upon request from the Mbarara University of Science and Technology Research Ethics Committee, via phone (+256780199188) or email (pro@must.ac.ug), for researchers who meet the criteria for access to confidential data.

**Funding:** The author(s) received no specific funding for this work.

**Competing interests:** The authors have declared that no competing interests exist.

## Conclusions

This study's primary finding is that fluid resuscitation in the emergency unit did not significantly increase or decrease three-day mortality for febrile children in Uganda. Universally aggressive or fluid-sparing emergency unit protocols are unlikely to be best practices, and choices about fluid resuscitation should be individualized.

## Introduction

Sepsis is a leading cause of mortality among children globally, with over 20 million cases in children under 5 leading to 2.9 million deaths in 2017 [1]. In low- and middle-income countries (LMICs), the case fatality rate for pediatric sepsis averages 31.7% [2]. Fluid resuscitation is a standard part of sepsis care in high-income countries. High-income guidelines call for aggressive fluid resuscitation in children with sepsis and septic shock, with a fluid bolus of 20mL/kg recommended for all patients with shock [3, 4]. Little, however, is known about fluid resuscitation in LMICs generally or sub-Saharan Africa specifically.

In 2011, a large randomized controlled trial, Fluid Expansion as Supportive Therapy (FEAST), was performed to find the optimal fluid resuscitation to reduce mortality in children aged 12 and under with severe infections in sub-Saharan Africa. This trial was stopped early for harm after the investigators unexpectedly found strong evidence that fluid boluses given to children with severe sepsis *increased* 48 hour mortality by 3.3% (Odds Ratio (OR) 1.45; 95% confidence interval [CI], 1.13 to 1.86) compared to children who received no fluid bolus [5]. This study generated a vigorous debate about generalizability of these results as conditions commonly receiving fluid resuscitation (including severe dehydration or hypovolemia) were excluded in FEAST [6–15].

A recent review on pediatric sepsis in both high- and low-income settings acknowledged the above controversy and lack of strong additional supporting or contradicting trial data [16]. Similarly a systematic review of pediatric sepsis (explicitly excluding tropical infections like malaria and dengue) found existing studies heterogeneous and of small sample size, precluding meta-analysis [17]. One small pilot trial from the UK and a proposed Canadian protocol with unpublished data at the time of the writing of this manuscript are the only RCTs that have assessed the mortality impact of fluids in sepsis in high-income settings [18, 19]. This uncertainty has resulted in a clinical discordance in East Africa with studies suggesting that providers tend to adopt a fluid-sparing strategy in pediatric sepsis while organizational guidelines continue to promote initial fluid bolus therapy [20–22].

The uncertainty around initial fluid resuscitation is particularly salient for emergency care clinicians in Africa. The increasing adoption of specialized emergency care in sub-Saharan Africa will further transition fluid resuscitation in pediatric sepsis to the emergency unit setting. However, no RCTs have been published to date to specifically investigate the mortality effect of fluid expansion therapy in sub-Saharan African emergency units for pediatric patients. Furthermore, fluid resuscitation in the emergency department is often required for undifferentiated patients. Febrile pediatric emergencies are often complicated by the presence of vomiting, diarrhea, poisonings, dehydration, and traumatic injury—all of which were excluded from the FEAST trial.

The World Health Organization has identified improving sepsis care as an important target in resolution WHA 70.7 and a recent global report [23, 24]. Current guidelines for EM in Africa have therefore defaulted to using traditional World Health Organization recommendations despite a lack of evidence base to clearly describe whether this approach will impact

mortality. Given the ongoing uncertainty, the objective of this manuscript is to test the hypothesis that administering a fluid bolus to febrile children in an East African emergency unit would increase mortality, similar to the effect seen in the FEAST trial.

## Materials and methods

This study investigates pediatric mortality associated with fluid bolus in a sub-Saharan African emergency unit in rural Uganda using existing longitudinal data from 2012–2019 for retrospective cohort analysis, using propensity score matching and logistic regression.

### Study setting

Global Emergency Care (GEC) is a U.S.- and Ugandan-based non-governmental organization founded in 2008 which provides emergency care training in Uganda. In collaboration with Karoli Lwanga Hospital in Rukungiri, Uganda, GEC developed a 2-year emergency training program; graduates of the program provide emergency care in a dedicated emergency unit. Karoli Lwanga Hospital is in rural southwest Uganda. The clinical setting, resource availability, and training program are comprehensively described elsewhere [25, 26]. The annual pediatric census of the emergency unit has been relatively stable since 2009, with a mean of 1,260 pediatric patients aged 2 months to 12 years old seen annually [26]. Approximately 60% of those patients were admitted to an inpatient pediatric ward whose resources remained relatively unchanged over the course of the study period.

### Data collection

GEC has maintained a prospectively collected quality assurance database of all emergency unit visits since 2009, including data about demographics, chief complaints, vital signs, laboratory and radiology results, diagnoses, disposition, and patient outcomes since 2009. Data about treatments given and procedures performed in the emergency unit have been recorded since 2012. The six-bed emergency unit was staffed by at least one (and typically two) clinicians and at least one nurse and one nursing student during the entire operating hours from 0800–2400. Additionally, a trained research assistant was present in the emergency unit during all operating hours who input the paper charts into the electronic database while the patient was still in the emergency unit. This staffing situation allowed for a very high fidelity of data capture. Digital scans were then made of every chart to allow review as needed.

Follow-up was done for all admitted and discharged patients at three-days to establish mortality outcomes for emergency unit visits. Patients that were discharged were contacted via phone on day three and if a patient could not be reached on the initial attempt, calls were made daily for seven consecutive days before they were considered as "lost to follow-up". Data was imported, cleaned, and analyzed in Stata Statistical Software version 16.1 (StataCorp, College Station, TX), and was de-identified and abstracted for analysis by a single researcher (BR). Ethics review board approval and waiver of patient consent was provided by the Mbarara University of Science and Technology Institutional Review Committee (No. 11/08-12) and University of Massachusetts Institutional Review Board (#14570).

### Subject selection

All visits between March 2012 and December 2019 were included in analysis for children who were (1) aged $\geq$ two months and $\leq$ 12 years; (2) had an abnormal body temperature ($\geq$37.5˚C or <36˚C); (3) seen in the Karoli Lwanga Hospital emergency unit and were admitted (either to the ward or directly to the operating theatre) or died in the emergency

unit; (4) had complete demographic information; (5) had complete follow-up data. Temperature cutoffs were taken from the FEAST protocol. Discharged patients were excluded (n = 752) from analysis because they had far lower rates of both the intervention (fluids) and the outcome (death).

## Data analysis

**Propensity score matching.** The association between IV fluid resuscitation and mortality is likely to be "confounded by indication" with more severely ill children both more likely to receive the intervention (IV fluids) and to have the outcome (death). Confounding by indication is typically addressed with regression adjustment, which may be inadequate if exchangeable counterparts do not exist [27]. To address this challenge, propensity score matching was performed. We used *psmatch2* with a bootstrap method to estimate standard error. We assigned a dependent variable of IV fluids and independent variables of history of vomiting and/or diarrhea, age-adjusted tachycardia, clinician impression of clinical condition, and age. These variables were informed by interviews with both the clinicians who treated pediatric patients and those who developed training curriculum and protocols.

**Missing data handling.** Missing data were imputed using multiple imputation via the *mi impute* command. Imputation was done for categorical variables of hypoxia, tachypnea, severe tachycardia, and hypothermia (as defined above) with death as an auxiliary variable correlated with missingness. Imputation used chained equations over ten imputations. Imputation was performed on matched data after matching had been completed, and patients were not matched based on imputed data.

**Logistic regression analysis.** The primary outcome of three-day mortality was modeled using multivariable logistic regression. All available patient characteristics (described in Table 1 below) were included in univariate analysis, and all those that had a univariate p-value $\leq 0.10$ were included in the multivariable model. Variables included in the model were age group (one year or younger, between 1 and 5 years old, 5 years and older), HIV status, hypoxia ($SpO_2 < 92\%$), tachypnea (maximum respiration rate $\geq 60$ if aged $< 2$ months, $\geq 50$ if aged 2 months to 1 year, $\geq 40$ if aged 1 to 5 years, $\geq 30$ if aged $>5$ and $\leq 12$ years) severe tachycardia (heart rate $\geq 180$ beats per minute [bpm] if aged one year or younger, $\geq 160$ bpm if aged between 1 and 5 years, $\geq 140$ if aged 5 years or greater), hypothermia ($< 36°C$) instead of hyperthermia ($\geq 37.5°C$). Malaria and presence of vomiting and/or diarrhea were explicitly excluded for failing to meet the univariate cutoff for significance. Fluid administration in the emergency unit was added as a final variable to the model to test its independent association with mortality. Complete case analysis and multiple imputation analysis were performed on both the matched and unmatched datasets. As a sensitivity analysis, the model was also applied to the subset of patients that met the inclusion criteria in the FEAST trial: (1) aged $\geq$ two months and $\leq 12$ years (2) abnormal temperature or complaint of fever; (3) impaired consciousness and/or respiratory distress; (4) impaired perfusion including one or more of: weak pulse, delayed capillary refill time, severe tachycardia, (5) absence of severe malnutrition, gastroenteritis, noninfectious causes of shock (e.g., trauma, surgery, or burns), and conditions for which volume expansion is contraindicated.

**Power calculation.** Post-hoc power calculations were performed with 80% power and an alpha = 0.05. The pre-matched data (n = 1561 no fluids, n = 1526 fluids, mortality = 4.0%) had power to detect a mortality change (risk difference) of 2.2% and an OR = 1.61. The propensity score matched data (n = 1180 fluids and no fluids, mortality = 4.1%) had power to detect a mortality change (risk difference) of 2.6% and an OR = 1.66.

**Table 1. Patient characteristics pre- and post-propensity score matching.**

| | UNMATCHED DATA | | | MATCHED DATA | | |
|---|---|---|---|---|---|---|
| | No Fluids | Fluids | p-value | No Fluids | Fluids | p-value |
| | (n = 1561) | (n = 1526) | | (n = 1180) | (n = 1180) | |
| | n (%) | n (%) | | n (%) | n (%) | |
| Age group, n (%) | | | | | | |
| Infant | 332 (21.3%) | 313 (20.5%) | 0.416 | 271 (23.0%) | 294 (24.9%) | 0.388 |
| 1–5 yrs | 939 (60.2%) | 901 (59.0%) | | 668 (56.6%) | 666 (56.4%) | |
| 5–12 yrs | 290 (18.6%) | 312 (20.4%) | | 241 (20.4%) | 220 (18.6%) | |
| Female | 692 (44.3%) | 665 (43.6%) | 0.674 | 508 (43.1%) | 510 (43.2%) | 0.934 |
| HIV | 30 (1.9%) | 22 (1.4%) | 0.300 | 29 (2.5%) | 18 (1.5%) | 0.105 |
| Malaria | 381 (24.4%) | 305 (20.0%) | 0.003 | 309 (26.2%) | 235 (19.9%) | <0.001 |
| Vomiting and/or diarrhea | 314 (20.1%) | 509 (33.4%) | <0.001 | 311 (26.4%) | 299 (25.3%) | 0.573 |
| Heart Rate | | | | | | |
| Severe Tachycardia | 477 (30.6%) | 667 (43.7%) | <0.001 | 438 (37.1%) | 468 (39.7%) | 0.204 |
| Missing | 15 (1.0%) | 7 (0.5%) | | 0 (0.0%) | 0 (0.0%) | |
| Oxygen saturation | | | | | | |
| Hypoxia | 376 (24.1%) | 345 (22.6%) | 0.294 | 292 (24.7%) | 262 (22.2%) | 0.198 |
| Missing | 46 (2.9%) | 35 (2.3%) | | 33 (2.8%) | 26 (2.2%) | |
| Respiratory rate | | | | | | |
| Tachypnea | 312 (20.0%) | 341 (22.3%) | 0.255 | 260 (22.0%) | 244 (20.7%) | 0.722 |
| Missing | 47 (3.0%) | 41 (2.7%) | | 31 (2.6%) | 31 (2.6%) | |
| Hypothermia (instead of fever) | | | | | | |
| Yes | 646 (41.4%) | 460 (30.1%) | <0.001 | 393 (33.3%) | 419 (35.5%) | 0.260 |
| Clinical impression | | | | | | |
| "Not sick" | 512 (32.8%) | 375 (24.6%) | <0.001 | 293 (24.8%) | 317 (26.9%) | 0.525 |
| "Sick" | 975 (62.5%) | 1068 (70.0%) | | 825 (69.9%) | 804 (68.1%) | |
| "Toxic" | 74 (4.7%) | 83 (5.4%) | | 62 (5.3%) | 59 (5.0%) | |
| Mortality | 66 (4.2%) | 57 (3.7%) | 0.484 | 48 (4.1%) | 48 (4.1%) | 1.000 |

All p-values calculated using chi-squared

## Results and discussion

### Results

**Study population.** Between Mar 24, 2012–Dec 31, 2019, there were 6925 emergency unit visits for patients aged between 2 months and twelve years that had complete follow up (Fig 1). Of those, 3,087 patients met inclusion and exclusion criteria listed in Methods above.

Patient characteristics are presented in Table 1 below. Significant differences between the "Fluids" and "No Fluids" cohorts existed prior to matching, including age group, malaria, presence of vomiting and/or diarrhea, severe tachycardia, hypothermia (instead of hyperthermia), and clinical impression of disease severity. Post-matching, all characteristics were balanced except for malaria. Graphical representation of covariate balance with matching is presented in Supporting Information (S1A and S1B Fig).

Total all-cause mortality across the unmatched data set was 4.0%. The difference in unadjusted mortality for children receiving fluid (n = 57, 3.7%, 95%CI 3.3%–5.3%) and those who did not (n = 66, 4.2%, 95%CI 2.8%–4.8%) was not statistically significant (p = 0.484). Within the matched data sets, overall mortality was 4.1%. The difference in unadjusted mortality

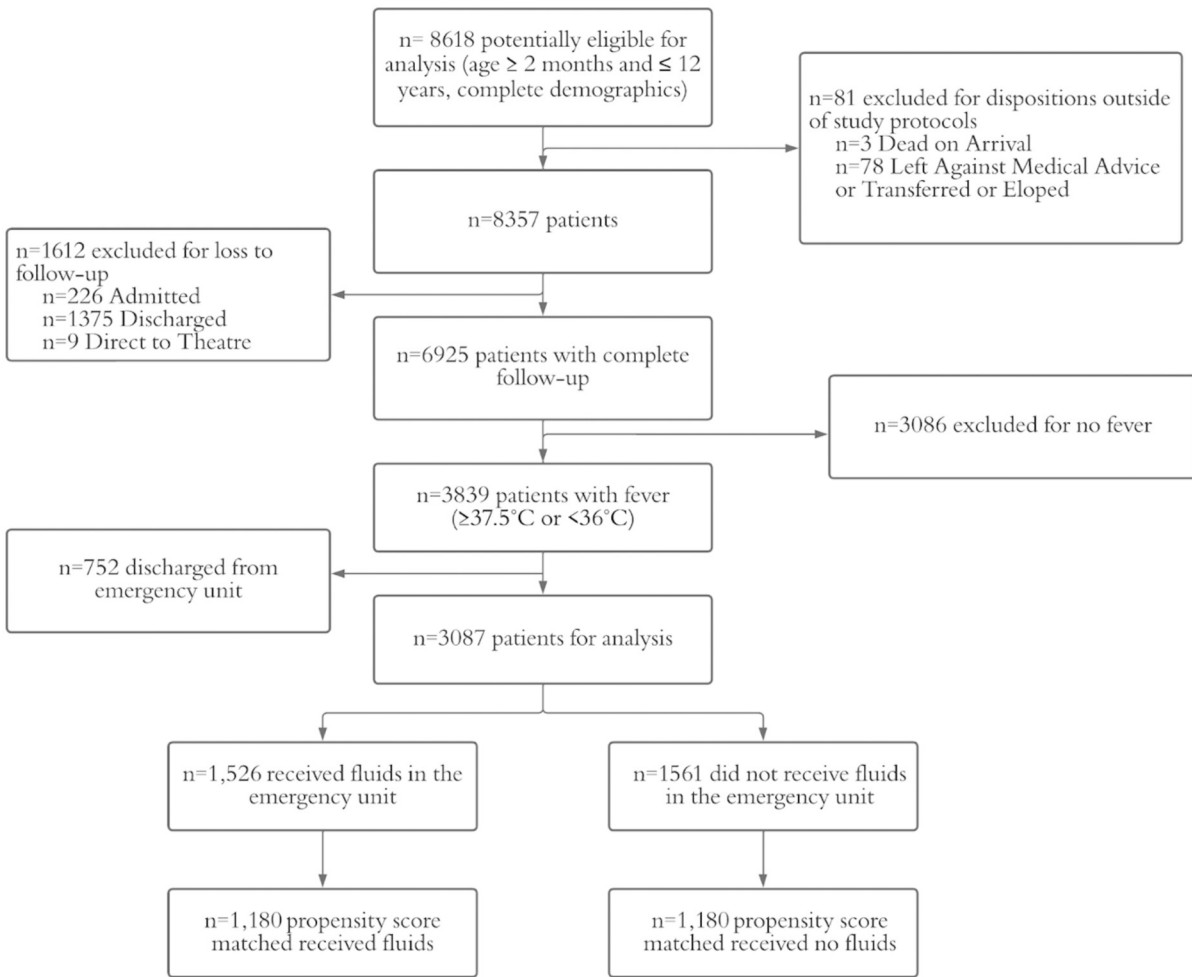

**Fig 1. Patient flow.** Graphical representation of inclusion and exclusion criteria for the study. Patients meeting age and temperature criteria and with available follow-up were assessed for fluid administration. Patients from each cohort that did and did not receive fluids were propensity score matched for their likelihood to receive fluids.

between children receiving fluids (n = 48, 4.1%, 95%CI 3.0%–5.4%) and those who did not (n = 48, 4.1%, 95%CI 3.0%–5.4%) was not statistically significant (p = 1.000).

Presence of missing data was low (≤ 3.0% across all variables) and similar between cohorts. Fluid dosing was available in 98.1% of patients that received fluids. Weights were only recorded in 50.4% of patients that received fluids, and 47.1% of patients that did not. Using the available data, the mean dose of fluids was 25.5 mL/kg (standard deviation 19.9). Of patients that got fluids, 1050 received normal saline, 462 received lactated Ringer's solution, and 12 received a mix of both. Looking at the patients that got fluids and had a recorded weight, 22.1% received less than 20 cc/kg (n = 170), 60.6% (n = 466) received between 20–29 cc/kg, and 17.3% (n = 133) received 30 or more cc/kg. The mortality within each of those three groups was 4.7% (n = 8), 3.9% (n = 18) and 5.3% (n = 7) and was not statistically significantly different (p = 0.7). Further analysis using weight-based dosing was not attempted due to the high rate of missingness.

A multivariable logistic regression model for mortality was developed to control for confounders (abnormal vital signs, clinical impression of severity, HIV coinfection, age, gender)

**Table 2. Unmatched and matched data logistic regression analysis: Complete case analysis and multiple imputation.**

| | Unmatched Data | | | | Matched Data | | | |
|---|---|---|---|---|---|---|---|---|
| | Complete Case (n = 2920) | | Multiple Imputation (n = 3087) | | Complete Case (n = 2241) | | Multiple Imputation (n = 2360) | |
| | Odds Ratio [95% CI] | p-Value | Odds Ratio [95% CI] | p-Value | Odds Ratio [95% CI] | p-Value | Odds Ratio [95% CI] | p-Value |
| Age Group | | | | | | | | |
| Infant <1 | 1.14 [0.68–1.90] | 0.617 | 1.39 [0.87–2.23] | 0.17 | 1.09 [0.61–1.95] | 0.761 | 1.34 [0.79–2.29] | 0.283 |
| Young child 1–5 | REF | | REF | | REF | | REF | |
| Older child 5–12 | 1.01 [0.58–1.77] | 0.968 | 1.04 [0.61–1.76] | 0.892 | 1.08 [0.58–2.00] | 0.81 | 1.08 [0.60–1.96] | 0.795 |
| HIV Status | | | | | | | | |
| Negative | REF | | REF | | REF | | REF | |
| Positive | 2.94 [1.03–8.38] | 0.043 | 2.60 [0.92–7.34] | 0.072 | 2.35 [0.73–7.57] | 0.152 | 2.10 [0.66–6.74] | 0.211 |
| Gender | | | | | | | | |
| Male | REF | | REF | | REF | | REF | |
| Female | 1.35 [0.89–2.04] | 0.161 | 1.41 [0.95–2.08] | 0.089 | 1.36 [0.85–2.18] | 0.201 | 1.43 [0.91–2.24] | 0.117 |
| Oxygen Sat # Resp Rate | | | | | | | | |
| Normal O2# Normal rate | REF | | REF | | REF | | REF | |
| Normal O2#Tachypnea | 2.56 [1.29–5.10] | 0.007 | 2.44 [1.25–4.77] | 0.009 | 3.18 [1.48–6.84] | 0.003 | 2.96 [1.39–6.29] | 0.005 |
| Hypoxia # Normal rate | 3.43 [1.97–5.95] | <0.001 | 3.39 [1.97–5.84] | <0.001 | 3.99 [2.09–7.58] | <0.001 | 3.92 [2.06–7.47] | <0.001 |
| Hypoxia # Tachypnea | 4.50 [2.49–8.13] | <0.001 | 4.51 [2.56–7.95] | <0.001 | 5.46 [2.76–10.8] | <0.001 | 5.66 [2.97–10.8] | <0.001 |
| Heart Rate | | | | | | | | |
| No severe tachycardia | REF | | REF | | REF | | REF | |
| Severe tachycardia | 1.58 [0.99–2.51] | 0.053 | 1.47 [0.95–2.28] | 0.084 | 1.58 [0.94–2.67] | 0.084 | 1.43 [0.88–2.34] | 0.153 |
| Temperature | | | | | | | | |
| Hyperthermic | REF | | REF | | REF | | REF | |
| Hypothermic | 4.19 [2.65–6.63] | <0.001 | 3.96 [2.57–6.11] | <0.001 | 3.82 [2.27–6.46] | <0.001 | 3.63 [2.21–5.96] | <0.001 |
| Clinical Condition | | | | | | | | |
| "Not sick" | REF | | REF | | REF | | REF | |
| "Sick" | 2.63 [1.22–5.70] | 0.014 | 2.68 [1.30–5.55] | 0.008 | 2.54 [0.98–6.6] | 0.056 | 2.91 [1.13–7.5] | 0.027 |
| "Toxic" | 14.5 [6.21–33.9] | <0.001 | 16.3 [7.4–35.9] | <0.001 | 15.2 [5.39–42.6] | <0.001 | 19.0 [6.91–52.0] | <0.001 |
| Fluid Bolus | | | | | | | | |
| Not given | REF | | REF | | REF | | REF | |
| Given in emergency unit | 0.93 [0.61–1.42] | 0.736 | 0.94 [0.63–1.39] | 0.741 | 1.06 [0.66–1.71] | 0.8 | 1.09 [0.69–1.70] | 0.718 |

and identify the independent contribution of fluid administration to mortality. Four models were analyzed looking at complete case analysis and multiple imputation in both matched and unmatched datasets and are presented in Table 2. Model calibration and discrimination for complete case analysis was similar and excellent in both the unmatched model (AUROC = 0.84, Brier score = 0.031 and Hosmer-Lemeshow p-value = 0.76) and the matched model (AUROC = 0.85, Brier score = 0.031, and Hosmer-Lemeshow p-value = 0.91). Model calibration and discrimination for multiple imputation analysis was similar and excellent in both the unmatched dataset (AUROC = 0.84, Brier score = 0.034 and Hosmer-Lemeshow p-value = 0.99) and the matched dataset (AUROC = 0.84, Brier score = 0.033, and Hosmer-Lemeshow p-value = 1.00).

Complete case analysis of unmatched data required exclusion of 167 of the 3,087 total patients for missing data. That subset of excluded patients had a significantly higher mortality rate as compared to those included in the model (9.0% [n = 15] vs. 3.7% [n = 108], p = 0.001). Complete case analysis of the matched dataset required exclusion of 119 of 2,360 total patients, and that subset also had a significantly higher mortality than those included in the model

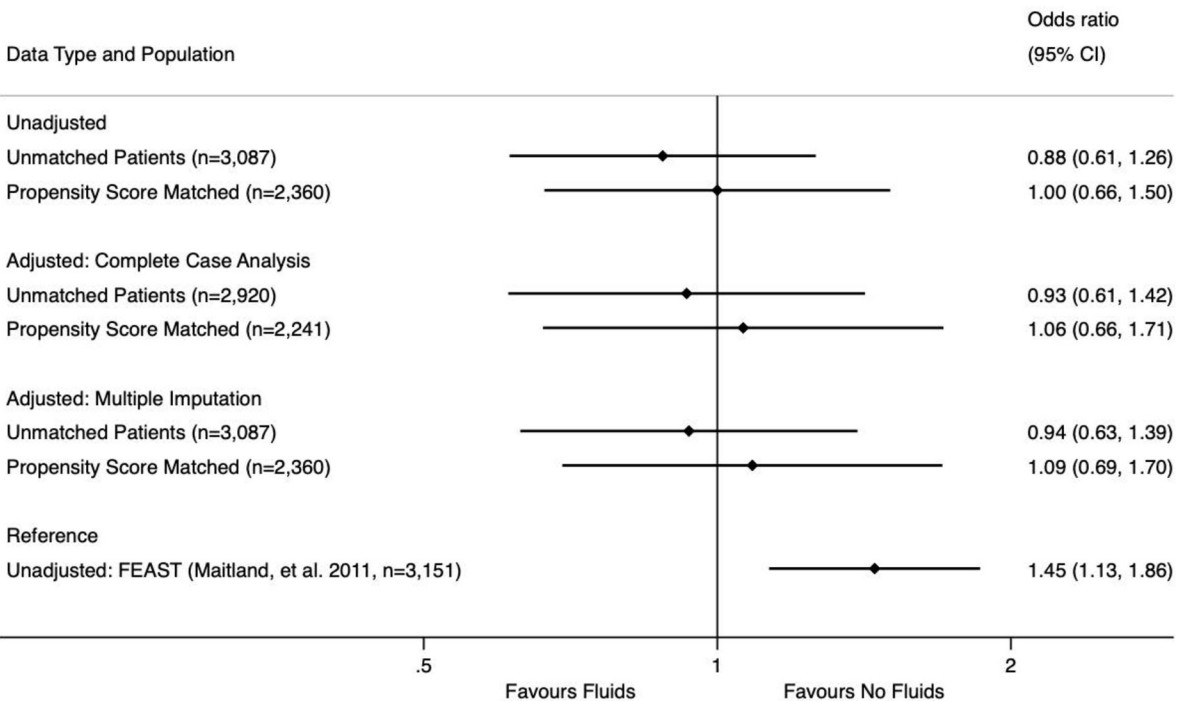

**Fig 2. Adjusted and unadjusted odds ratios for mortality with fluid administration.** Graphical representation of the odds ratios and 95% confidence intervals for the mortality associated with fluid administration in six different groups: unadjusted (univariate) analysis for (1) unmatched and (2) propensity score matched patients; multivariable logistic regression complete case analysis for (3) unmatched and (4) propensity score matched patients; multivariable logistic regression multiple imputation analysis for (5) unmatched and (6) propensity score matched patients. The unadjusted (univariate) analysis from the seminal FEAST trial (Maitland, et al. 2011) is included for reference.

(9.2% [n = 11] vs. 3.8% [n = 85], p = 0.003). Multiple imputation allowed inclusion of those higher risk patients in subsequent analysis of both matched and unmatched data.

In all four logistic regression models, fluids were *not found to be significantly associated with increased or decreased mortality*. To visually summarize our analysis, the unadjusted mortality in both matched and unmatched data is presented as OR alongside the adjusted mortality from the four logistic regression models (complete case analysis and multiple imputation analysis of unmatched and matched datasets) and the unmatched mortality reported in the FEAST trial as Fig 2.

As sensitivity analysis, the subset of patients that met FEAST inclusion and exclusion criteria (n = 468, 15.1% of patients overall) were analyzed and matched separately. There was no statistically significant mortality benefit or penalty for fluids either in the unmatched dataset (No fluids: 13 deaths in 192 patients, 6.8%, Fluids: 10 deaths in 276 patients, 3.6%, p = 0.12) or in the matched dataset (No fluids: 12 deaths in 186 patients, 6.5%, Fluids: 7 deaths in 182 patients, 3.9%, p = 0.26). We then performed logistic regression on both datasets including complete case analysis and multiple imputation. No significant mortality benefit or penalty was seen, and those results are summarized in Fig 3.

## Discussion

This study's primary finding is that there was no statistically significant association between fluid resuscitation in the emergency unit and mortality for febrile children aged two months to 12 years. Notably, these emergency unit patients were undifferentiated and not required to meet the exclusion (e.g., no evidence of malnutrition, nausea/vomiting, trauma) or inclusion

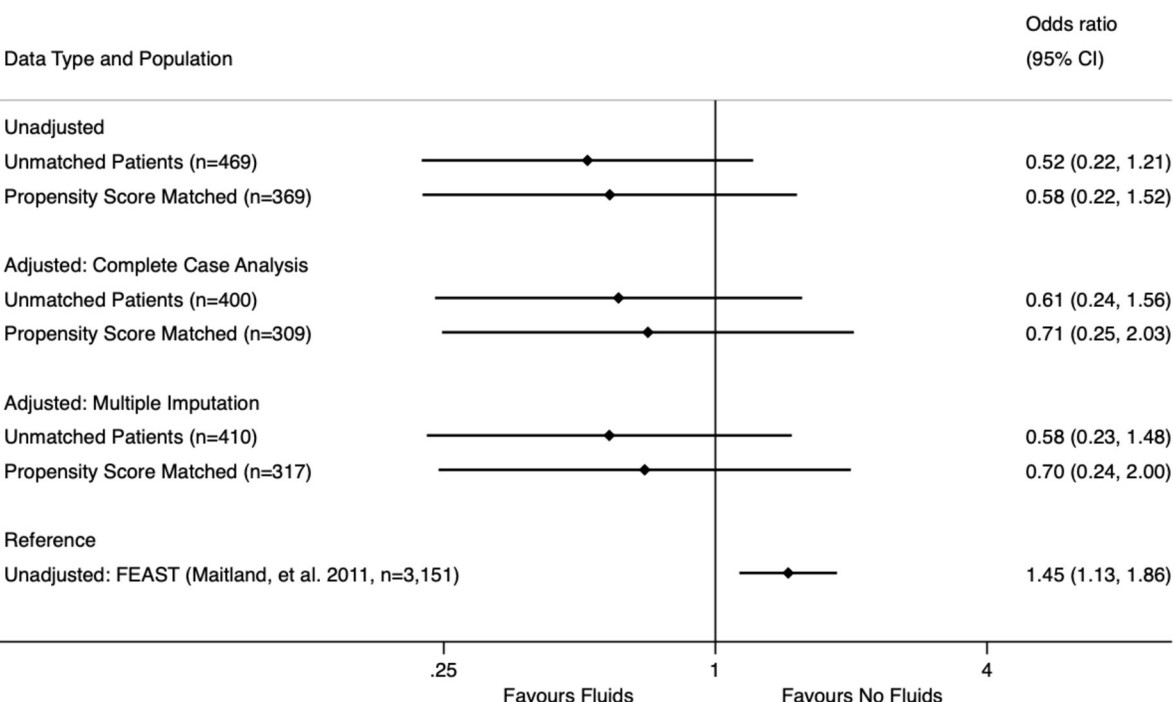

**Fig 3. Subgroup mortality analysis for patients meeting FEAST inclusion/exclusion criteria.** Graphical representation of the odds ratios and 95% confidence intervals for the mortality associated with fluid administration in six different groups of patients that meet the inclusion and exclusion criteria for the seminal FEAST trial (Maitland, et al. 2011): unadjusted (univariate) analysis for (1) unmatched and (2) propensity score matched patients; multivariable logistic regression complete case analysis for (3) unmatched and (4) propensity score matched patients; multivariable logistic regression multiple imputation analysis for (5) unmatched and (6) propensity score matched patients. The unadjusted (univariate) analysis from the FEAST trial is included for reference.

criteria (e.g. signs of impaired perfusion, severely deranged vitals) of the FEAST trial. This more accurately reflects the range of undifferentiated patients who receive emergency care. The strength of this null finding was supported by multiple analytic approaches—including propensity score matching, multiple imputation and logistic regression—which all failed to show any significant association between fluids and mortality. The choice of analytic methods optimized the natural experiment provided by this retrospective database to more closely resemble trial methodology. We produced four models with excellent calibration and discrimination for pediatric mortality in a rural sub-Saharan African emergency unit, and all showed no association between fluids and an increase or decrease in mortality (OR = 0.93–1.09). To our knowledge, no similar model(s) of the impact of fluids on emergency unit pediatric mortality have been published to date.

Sensitivity analysis looking at the subset of patients that did fit FEAST exclusion and inclusion criteria also showed no significant mortality penalty or benefit. Moreover, all four models showed trends towards *reduced* mortality with fluids (mean OR ranged between 0.52 and 0.70). This was somewhat surprising as our a priori assumption was that these patients were more likely to show a trend towards harm from fluids. However, the wide confidence intervals in these models limits further interpretation.

This study was designed to see if the increased mortality associated with fluid boluses for pediatric inpatients with severe infections in the FEAST trial would also be seen for fluid boluses in the lower mortality, more heterogeneous emergency unit population of febrile pediatric patients. The total number of patients in our study (n = 3,087) was almost identical to the FEAST trial (n = 3,151). Our power calculations indicate that unmatched and matched

analyses were both adequately powered to have seen the 3.3% *absolute* mortality increase reported in FEAST. This power allows us to conclude that this absolute mortality penalty does not exist for fluid boluses in febrile emergency unit children. Our analyses, however, were underpowered to detect the *relative* increase in mortality (OR = 1.44) seen in FEAST. Despite the relatively large number of patients in our study, the mortality rate in our emergency unit data—even after excluding discharged patients who had zero deaths in 752 patients—was far lower than that seen in the FEAST trial (4.0% vs. 9.5%). Even in the 15% of emergency unit patients meeting FEAST criteria, the mortality was far lower (4.9%).

This mortality difference may be due to several factors. Baseline childhood mortality rates have decreased in Uganda from 82.1 to 45.8/1,000 live births from 2009 (when the FEAST trial began enrollment) to 2019 (when data collection stopped in our study) [28]. Other possibilities include different care seeking behavior for this rural emergency unit than in the urban hospital inpatient settings used to enroll patients in FEAST. It is also possible that emergency unit patients received different benefits or harms from fluids because of the earlier timing for emergency unit fluid resuscitation as compared to inpatient resuscitation. Regardless of the ultimate cause for the lower mortality rates, the low number of deaths meant that our unmatched data was only powered to see OR = 1.61 and our matched data was powered to see OR = 1.66. Though our analyses did not show a trend towards the harm seen in FEAST (in fact, it demonstrated a trend toward benefit amongst patients meeting FEAST criteria), the lack of power prevents us from drawing strong, final conclusions about the relative harm or benefit.

FEAST was and remains a seminal trial that provided strong evidence for the judicious use of fluid resuscitation in severely ill children in the inpatient setting. However, clinicians working in Ugandan emergency units are faced with undifferentiated febrile pediatric patients who may not be critically ill and who often have comorbidities of malnutrition, clinical dehydration, nausea/vomiting, and trauma. Clinicians currently lack evidence-based guidance for optimal fluid resuscitation in these patients. Our study suggests that fluid boluses are not significantly associated with mortality in a sub-Saharan African emergency unit setting. Providing guidance for emergency clinicians has become a priority as emergency medicine develops under the guidance of the WHO throughout sub-Saharan Africa and in low-income countries more generally [29]. While future randomized controlled trials may ideally identify populations that receive benefit from emergency unit fluid resuscitation, our study provides evidence that there is no class mortality effect of emergency unit fluids on undifferentiated, lower risk febrile pediatric patients. Therefore, neither universally fluid-sparing nor aggressive resuscitation protocols for febrile pediatric patients are best practice. Emergency care clinicians and policy-makers should continue to emphasize individualized care when it comes to fluids for pediatric patients.

## Limitations

Several limitations to the current study must be noted. As discussed thoroughly in Methods and Discussion, the study was underpowered due to the low overall mortality rate, despite eight years of data collection. Given the retrospective nature of the data, the analysis was limited by available variables. Most notably, while fluid volume was recorded, patient weight was not routinely recorded and thus analysis accounting for weight-based dosing for fluid expansion was impossible. Because of the wide range of children sizes in the study, the inability to calculate weight-based dosing meant that volume of fluid given, which has significant potential to impact mortality, could not be included in the study. Furthermore, because rate of administration was not recorded in this data the analysis was unable to address the contribution of rapid (bolus) versus continuous (maintenance) fluid administration to mortality. Matching

helped balance disparities between cohorts for all variables except for malaria. This limitation was mitigated by the fact that malaria was not independently associated with mortality and fluids and did not appear in the final models. Data regarding inpatient fluid management was not available and the mortality impact of this fluid management after admission was outside the scope of this study. However, as all patients were admitted to the same pediatric team, the impact of inpatient care was felt to equally affect both arms of the study. Mortality was recorded at day three following admission. While this metric was chosen deliberately to best represent the impact of emergency unit care, shorter (24 hour) and longer (1–4 week) mortality rates may provide insights this metric could not. Finally, this study is limited to the findings from a single site in rural Uganda. The generalizability of these findings is unclear in the absence of similar analysis from other emergency units in Uganda and throughout sub-Saharan Africa.

## Conclusions

In this study, we used methodology including propensity score matching and multiple imputation coupled with multivariable logistic regression modeling to provide the most accurate retrospective analysis possible of the mortality impact of fluid boluses on an undifferentiated group of febrile pediatric patients in a Ugandan emergency unit from 2012–2019. Both crude and adjusted mortality analysis showed no significant association between fluid boluses given in the emergency unit and mortality. The lack of a class effect of fluids suggests that neither universal fluid-sparing nor aggressive resuscitation protocols for febrile pediatric emergency patients are best practice and the choice of fluids should continue to be individualized.

## Supporting information

**S1 Fig. Matching support and variable balance after matching.** (A) Graphical representation of patients that are on and off support for propensity score matching using psmatch2 in Stata 16. (B) Graphical representation of standardized bias in the variables included in propensity score matching before and after matching using psmatch2 in Stata 16.
(TIF)

## Acknowledgments

The authors would like to thank the clinicians who have provided the life-saving care in the emergency unit at Karoli Lwanga Hospital. Their impact on the lives of their patients and their community cannot be overstated. We would also like to thank Dr. Michael Kohn and Dr. Holly Elser for their significant methodological contributions to our analysis and matching efforts. Finally, we wish to thank the administrators and physicians at Karoli Lwanga Hospital for their support of ongoing clinical and research efforts. The Global Emergency Care Investigator Group is comprised of Mark Bisanzo, Heather Hammerstedt, Brad Dreifuss, and Stacey Chamberlain.

## Author Contributions

**Conceptualization:** Brian Rice, Serena Nakato.

**Data curation:** Brian Rice.

**Formal analysis:** Brian Rice, Jessica Hawkins, Serena Nakato.

**Investigation:** Brian Rice, Jessica Hawkins.

**Methodology:** Brian Rice.

**Project administration:** Brian Rice, Nicholas Kamara.

**Supervision:** Brian Rice.

**Validation:** Brian Rice.

**Visualization:** Brian Rice, Jessica Hawkins.

**Writing – original draft:** Brian Rice, Jessica Hawkins.

**Writing – review & editing:** Brian Rice, Jessica Hawkins, Serena Nakato, Nicholas Kamara.

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
