## [Decision Letter · Decision Letter 0]

24 May 2023

PONE-D-23-05026Mortality after emergency unit fluid bolus in febrile Ugandan childrenPLOS ONE

Dear Dr. Rice,

Thank you for submitting your manuscript to PLOS ONE. After careful consideration, we feel that it has merit but does not fully meet PLOS ONE’s publication criteria as it currently stands. Therefore, we invite you to submit a revised version of the manuscript that addresses the points raised during the review process.

Overall, the reviewers appreciated the rigor and importance of this work and believe the work will be of interest to the field. As this study addresses an important clinical question that has been subject to considerable debate, it will be important to address as many reviewer points as possible in the revised manuscript. I strongly encourage you to review their comments and incorporate any additional supportive analysis possible and to incorporate the limitations mentioned in a revised manuscript. Thank you for the submission.  Please submit your revised manuscript by Jul 08 2023 11:59PM. If you will need more time than this to complete your revisions, please reply to this message or contact the journal office at plosone@plos.org. Please include the following items when submitting your revised manuscript:A rebuttal letter that responds to each point raised by the academic editor and reviewer(s). You should upload this letter as a separate file labeled 'Response to Reviewers'.A marked-up copy of your manuscript that highlights changes made to the original version. You should upload this as a separate file labeled 'Revised Manuscript with Track Changes'.An unmarked version of your revised paper without tracked changes. You should upload this as a separate file labeled 'Manuscript'.

We look forward to receiving your revised manuscript.

Kind regards,

Andrea L. Conroy, PhD

Academic Editor

PLOS ONE

Journal Requirements:

Reviewers' comments:

Reviewer's Responses to Questions

**Comments to the Author**

1. Is the manuscript technically sound, and do the data support the conclusions?

Reviewer #1: Yes

Reviewer #2: Yes

2. Has the statistical analysis been performed appropriately and rigorously? 

Reviewer #1: Yes

Reviewer #2: Yes

3. Have the authors made all data underlying the findings in their manuscript fully available?

Reviewer #1: No

Reviewer #2: Yes

4. Is the manuscript presented in an intelligible fashion and written in standard English?

Reviewer #1: Yes

Reviewer #2: Yes

5. Review Comments to the Author

Reviewer #1: Mortality after emergency unit fluid bolus in febrile Ugandan children

The authors performed a retrospective study evaluating the association of emergency room fluid resuscitation with mortality among febrile children in Uganda. Propensity scoring was used to create matched cohorts, and a sensitivity analysis was done separately for patients meeting FEAST inclusion and exclusion criteria, a trial that was stopped early because of the increased mortality. There were just over 1500 in each of the matched groups (fluid vs. no fluid). There was no association of fluid resuscitation with mortality. There was also no difference in the sensitivity analysis. The authors conclude that extremes of under and over resuscitation may not be best practice and that clinical decisions should be individualized.

This is an important manuscript, even though underpowered given the lower mortality rate compared to FEAST. I think the authors highlight the need to individualize care. Overall, I am impressed by this work, and feel like it could be strengthened looking at mortality further from 3 day (i.e. 28 days) and other resource utilization (for example how long these children stayed in the hospital and what other therapies were provided, including mechanical ventilation). I have provided some major and minor comments below.

Major

1. The specific inclusion and exclusion criteria should be state din the methods (the exclusion of patients by not being admitted is currently in the results)

2. It seems strange to say that in MVR, fluids were not associated with increased or decreased mortality. This use of increased and decreased occurs in multiple places throughout the results and discussion, and it feels a bit redundant. Perhaps it would make more sense to say there was no association with mortality. I acknowledge that this doesn’t provide directionality, but I think it still conveys the message.

3. The authors suggest that looking at mortality further out (for example 28-day) may be helpful. Is there a study doing this, particularly for those admitted, and was there ongoing additional resuscitation. These are just general thoughts, but if this is being investigated, it would be worthwhile to state this.

Minor:

- There are several grammatical errors throughout.

- Line 251 – “There no…” it seems there is a word missing.

- On line 316 “our manuscript provides…..” This should be “our study provides”

Reviewer #2: This is a nice analysis that challenges/qualifies the findings of FEAST. The findings of the present, retrospective study, are, however, consistent with clinical “tradition” and experience: fluid resuscitation in the emergency unit is not harmful (and may even be helpful!). The FEAST methodology, is, of course, stronger, so this paper will not settle the debate. Still, these retrospective observational data seem important to provide countervailing observational data to the findings of the clinical trial. They may stimulate further inquiry into the mechanism by which fluids increased mortality, contrary to expectation, in FEAST. Furthermore, the “real world” setting for the present analysis may increase the confidence of clinicians wishing, by every impulse of their clinical reasoning, to provide fluid resuscitation in rural emergency departments in LMICs, that they are not likely doing harm.

Summary of the article:

Design: Retrospective cohort study (2012-2019) from a single centre (Karoli Lwanga, Rukungiri)

Exposure: fluid bolus versus no fluids (this is somewhat ill-defined -see below)

Outcome: Mortality at 3 days (telephone follow-up done)

Participants:

- Children 2 mo to 12 years of age, fever (or hypothermia), admitted

- Complete demographic info

- Complete follow-up data - as noted by the authors, death correlated with missingness; addressed (to the extent that this is possible) with multiple imputation)

Analysis:

Propensity score matched and unmatched analyses

Complete case analysis and multiple imputation analysis

Subgroup analysis of patients that met FEAST inclusion criteria

(great analysis, a strength of this study)

Comment (positive, no change needed):

The analysis (propensity score matching, multiple imputation for missing data, subgroup analysis of patients meeting inclusion criteria for FEAST) appears to be masterfully executed. This thorough analysis was indeed necessary, despite the “null” result, since these findings invite many questions in the shadow of FEAST.

I appreciated the transparent and critical analysis of the sample size (post hoc power). This clearly and completely addressed the question that arose as I read the manuscript about the power of a study with “null” conclusion (risk of type 2 statistical error). The power calculation indicated that the study was underpowered with respect to the relative increase in mortality seen in FEAST (but adequate to rule out the absolute mortality increase seen in FEAST), despite the large number of patients included over an 8 year period (though with only 57+66 events, 4% mortality).

Suggest including absolute risk difference in the abstract:

The authors should consider including the absolute risk difference with 95% confidence interval in the abstract (from at least one of the analyses), rather than simply stating “No significant mortality benefit or harm.” The upper and lower limit of the 95%CI of the mortality difference would help the reader understand the precision of the null finding (and judge whether a larger study is needed).

Comment (please address with a sentence or two in Discussion or Limitations)

Lacking are details of the fluid resuscitation, which may be important for explaining differences with FEAST. “Fluids” versus “no fluids” as a binary variable is a bit simplistic, given that there is a range of appropriate and inappropriate IV fluid therapies. The patient weight was only recorded in ~50%, so the mL/kg fluid dose could not be calculated for a large fraction of patients. The mean dose of fluids was 25.5 mL/kg (standard deviation 19.9), meaning that some patients received much more than 20mL/kg (recommended “bolus” fluid volume). The authors appropriately refrained from more in depth descriptive analyses of the fluid administration because of the limited quality of the data; however, these details may be important. The type of fluid was not specified (I could not find this in the paper) – normal saline? Ringer’s lactate? The frequency of IV fluid administration (or continuous infusion) was not specified. In a high-income setting, “maintenance” fluids are often administered after an initial bolus, in addition to fluids to correct the deficit at presentation. I acknowledge that, in the LMIC context, careful reassessment of fluid requirements may exceed nursing or medical capacity, but a single episode of fluid administration does not seem physiologic (frequent reassessment needed, these are critically ill patients). Was the IV fluid given once and never again? For critically ill patients, ongoing IV fluid input may be required before enteral feeds can be started. At what rate was the fluid given (“bolus” push, mL/hour)? This could be important if endothelial leak, pulmonary edema, and/or cerebral edema contribute to increased mortality with rapid fluid administration.

To address this limitation, I suggest that the authors add a sentence or two to the limitations paragraph, highlighting the details of the fluid therapy that would be helpful (e.g., in future studies) to better define the “exposure.”

Comment (no change needed)

Overall, the quality of the clinical data is impressive (presence of missing data was low, ≤ 3.0% across all variables). However, given that weight was only recorded in 50% (this is an important clinical parameter in pediatric medicine for dosing of all kinds of medications, not only fluids), one cannot help but harbour lingering doubts about the accuracy and thoroughness of ?paper records collected in a ED in a rural hospital in an LMIC for clinical purposes. Perhaps additional detail on the added human resources (“trained research assistants”) provided to the hospital over the years would be helpful to assuage some misgivings about the quality of retrospective clinical data. These data just seem surprisingly “clean” compared to other Ugandan hospital records I have worked with. The staff should be congratulated on such meticulous record keeping.

Using a “gestalt” of how sick the patient was, versus a standardized composite clinical severity score:

Clinical impression (“not sick,” “sick,” or “toxic”) was a highly prognostic variable. I agree that this assessment, when performed by a trained or experienced clinician, is valid. The authors should also consider using one of several published, standardized, validated composite severity scores such as LODS (works for non-malarial febrile illness – see for example Conroy AL, et al. Prospective validation of pediatric disease severity scores to predict mortality in Ugandan children presenting with malaria and non-malaria febrile illness. Crit Care. 2015; 19(1): 47), SICK (Signs of Inflammation in Children that Kill), or PEDIA. There are even composite severity scores derived from the FEAST cohort (FEAST Paediatric Emergency Triage (PET) Score). The purpose of using one of these would be to have an objective and standardized scale of how sick the patient was. Important to adjust effect of fluids on mortality for disease severity at baseline, which the score would do nicely. This is a discretionary revision.

Comment (no change needed)

The authors attempt to explain difference in mortality at the rural hospital (4%) compared to FEAST centres (9.5%): “Other possibilities include different care seeking behavior for this rural emergency unit than in the urban hospital inpatient settings used to enroll patients in FEAST.” I agree that, for example, Mulago Hospital in Kampala (a FEAST centre) is a national referral hospital and receives the sickest patients from across the country, including those that could not be managed at peripheral centres like Karoli Lwanga Hospital and many others. The patients are a select group of the sickest, that have previously failed standard management. The higher mortality in FEAST is not surprising to me.

6. PLOS authors have the option to publish the peer review history of their article (what does this mean?). If published, this will include your full peer review and any attached files.

Reviewer #1: No

Reviewer #2: No

---

## [Author Response · Author response to Decision Letter 0]

18 Jul 2023

Dear Reviewers and Editors,

We thank you for the opportunity to revise and resubmit our manuscript “Mortality after emergency unit fluid bolus in febrile Ugandan children”. We appreciate your close reading and excellent feedback and have addressed your comments below in a point-by-point manner.

Thank you again for your time and consideration.

Sincerely,

Brian Rice

General

1. PLOS ONE’s style requirements have been met for formatting.

2. A complete copy of PLOS’ questionnaire on inclusivity in global research has been attached.

3. The ethics statement in the Methods has been changed to “Ethics review board approval and waiver of patient consent was provided by the Mbarara University of Science and Technology Institutional Review Committee (No. 11/08-12) and University of Massachusetts Institutional Review Board (#14570).”

4. The same paragraph as #3 was added to expand our Ethics methods “Ethics review board approval and waiver of patient consent was provided by the Mbarara University of Science and Technology Institutional Review Committee (No. 11/08-12) and University of Massachusetts Institutional Review Board (#14570).”

Reviewer #1

Major

1. The specific inclusion and exclusion criteria should be state din the methods (the exclusion of patients by not being admitted is currently in the results)

a. RESPONSE: They have been moved to Methods, and Results changed to “Of those, 3,087 patients met inclusion and exclusion criteria listed in Methods above.”

2. It seems strange to say that in MVR, fluids were not associated with increased or decreased mortality. This use of increased and decreased occurs in multiple places throughout the results and discussion, and it feels a bit redundant. Perhaps it would make more sense to say there was no association with mortality. I acknowledge that this doesn’t provide directionality, but I think it still conveys the message.

a. RESPONSE: The statement “increased or decreased” has now been removed in all cases except the first time the result is described in “In all four logistic regression models, fluids were not found to be significantly associated with increased or decreased mortality”

3. The authors suggest that looking at mortality further out (for example 28-day) may be helpful. Is there a study doing this, particularly for those admitted, and was there ongoing additional resuscitation. These are just general thoughts, but if this is being investigated, it would be worthwhile to state this.

a. RESPONSE: We share the reviewer’s interest in longer term outcomes, but the QA database this data comes from was originally built to look only at short term outcomes and doesn’t contain 7 or 28-day outcomes.

Minor

1. Minor:

- There are several grammatical errors throughout.

- Line 251 – “There no…” it seems there is a word missing.

- On line 316 “our manuscript provides…..” This should be “our study provides”

a. RESPONSE: Line 251: changed to “There was no…”

b. RESPONSE: Line 316: “manuscript” has been changed to “study”

Reviewer #2

Major

1. Suggest including absolute risk difference in the abstract:

The authors should consider including the absolute risk difference with 95% confidence interval in the abstract (from at least one of the analyses), rather than simply stating “No significant mortality benefit or harm.” The upper and lower limit of the 95%CI of the mortality difference would help the reader understand the precision of the null finding (and judge whether a larger study is needed).

a. RESPONSE: The abstract has been edited to include this information.

2. Comment (please address with a sentence or two in Discussion or Limitations)

Lacking are details of the fluid resuscitation, which may be important for explaining differences with FEAST. “Fluids” versus “no fluids” as a binary variable is a bit simplistic, given that there is a range of appropriate and inappropriate IV fluid therapies. The patient weight was only recorded in ~50%, so the mL/kg fluid dose could not be calculated for a large fraction of patients. The mean dose of fluids was 25.5 mL/kg (standard deviation 19.9), meaning that some patients received much more than 20mL/kg (recommended “bolus” fluid volume). The authors appropriately refrained from more in depth descriptive analyses of the fluid administration because of the limited quality of the data; however, these details may be important. The type of fluid was not specified (I could not find this in the paper) – normal saline? Ringer’s lactate? The frequency of IV fluid administration (or continuous infusion) was not specified. In a high-income setting, “maintenance” fluids are often administered after an initial bolus, in addition to fluids to correct the deficit at presentation. I acknowledge that, in the LMIC context, careful reassessment of fluid requirements may exceed nursing or medical capacity, but a single episode of fluid administration does not seem physiologic (frequent reassessment needed, these are critically ill patients). Was the IV fluid given once and never again? For critically ill patients, ongoing IV fluid input may be required before enteral feeds can be started. At what rate was the fluid given (“bolus” push, mL/hour)? This could be important if endothelial leak, pulmonary edema, and/or cerebral edema contribute to increased mortality with rapid fluid administration.

To address this limitation, I suggest that the authors add a sentence or two to the limitations paragraph, highlighting the details of the fluid therapy that would be helpful (e.g., in future studies) to better define the “exposure.”

a. RESPONSE: Additional details were added into the results “Of patients that got fluids, 1050 received normal saline, 462 received lactated Ringer’s solution, and 12 received a mix of both. Looking at the patients that got fluids and had a recorded weight, 22.1% received less than 20 cc/kg (n=170), 60.6% (n=466) received between 20-29 cc/kg, and 17.3% (n=133) received 30 or more cc/kg. The mortality within each of those three groups was 4.7% (n=8), 3.9% (n=18) and 5.3% (n=7) and was not statistically significantly different (p=0.7).” 

We also added to the Limitations section “Furthermore, because rate of administration was not recorded in this data the analysis was unable to address the contribution of rapid (bolus) versus continuous (maintenance) fluid administration to mortality.”

3. Comment (no change needed)

Overall, the quality of the clinical data is impressive (presence of missing data was low, ≤ 3.0% across all variables). However, given that weight was only recorded in 50% (this is an important clinical parameter in pediatric medicine for dosing of all kinds of medications, not only fluids), one cannot help but harbour lingering doubts about the accuracy and thoroughness of ?paper records collected in a ED in a rural hospital in an LMIC for clinical purposes. Perhaps additional detail on the added human resources (“trained research assistants”) provided to the hospital over the years would be helpful to assuage some misgivings about the quality of retrospective clinical data. These data just seem surprisingly “clean” compared to other Ugandan hospital records I have worked with. The staff should be congratulated on such meticulous record keeping.

a. RESPONSE: We agree that the fidelity of the data is remarkable but would like to assure the reviewer that it is very genuine and something which a huge amount of effort and human resources were applied to over a decade. To help explain how this was accomplished we added the following to Methods “The six-bed emergency unit was staffed by at least one (and typically two) clinicians and at least one nurse and one nursing student during the entire operating hours from 0800 – 2400. Additionally, a trained research assistant was present in the emergency unit during all operating hours who input paper charts into the electronic database while the patient was still in the emergency unit. This staffing situation allowed for a very high fidelity of data capture. Digital scans were then made of every chart to allow review as needed.” I can also assure the reviewer that we had an additional layer of data quality as we scanned all the paper charts over ten years and had them for remote review along with in-country review of the hard copies. The author team has done multiple rounds of hand reviewing and spot checking with those scanned copies over the years and the data is actually of the reported quality.

4. Using a “gestalt” of how sick the patient was, versus a standardized composite clinical severity score: Clinical impression (“not sick,” “sick,” or “toxic”) was a highly prognostic variable. I agree that this assessment, when performed by a trained or experienced clinician, is valid. The authors should also consider using one of several published, standardized, validated composite severity scores such as LODS (works for non-malarial febrile illness – see for example Conroy AL, et al. Prospective validation of pediatric disease severity scores to predict mortality in Ugandan children presenting with malaria and non-malaria febrile illness. Crit Care. 2015; 19(1): 47), SICK (Signs of Inflammation in Children that Kill), or PEDIA. There are even composite severity scores derived from the FEAST cohort (FEAST Paediatric Emergency Triage (PET) Score). The purpose of using one of these would be to have an objective and standardized scale of how sick the patient was. Important to adjust effect of fluids on mortality for disease severity at baseline, which the score would do nicely. This is a discretionary revision.

a. RESPONSE: We would like to thank the reviewer for this suggestion. Looking at how non-clinician gestalt performs is something that has not been done for emergency medicine to our knowledge and is a very intriguing concept. Given that the current manuscript is already relatively methods heavy with several subgroup and sensitivity analyses we will look at a separate manuscript to pursue gestalt analysis in a robust manner.

5. Comment (positive, no change needed):

The analysis (propensity score matching, multiple imputation for missing data, subgroup analysis of patients meeting inclusion criteria for FEAST) appears to be masterfully executed. This thorough analysis was indeed necessary, despite the “null” result, since these findings invite many questions in the shadow of FEAST.

I appreciated the transparent and critical analysis of the sample size (post hoc power). This clearly and completely addressed the question that arose as I read the manuscript about the power of a study with “null” conclusion (risk of type 2 statistical error). The power calculation indicated that the study was underpowered with respect to the relative increase in mortality seen in FEAST (but adequate to rule out the absolute mortality increase seen in FEAST), despite the large number of patients included over an 8 year period (though with only 57+66 events, 4% mortality).

a. RESPONSE: Thank you so much for your kind comments.

6. Comment (no change needed)

The authors attempt to explain difference in mortality at the rural hospital (4%) compared to FEAST centres (9.5%): “Other possibilities include different care seeking behavior for this rural emergency unit than in the urban hospital inpatient settings used to enroll patients in FEAST.” I agree that, for example, Mulago Hospital in Kampala (a FEAST centre) is a national referral hospital and receives the sickest patients from across the country, including those that could not be managed at peripheral centres like Karoli Lwanga Hospital and many others. The patients are a select group of the sickest, that have previously failed standard management. The higher mortality in FEAST is not surprising to me.

a. RESPONSE: Thank you for your insight and understanding of the heterogeneity of care settings within Uganda.

---

## [Editor Report · Decision Letter 1]

16 Aug 2023

Mortality after emergency unit fluid bolus in febrile Ugandan children

PONE-D-23-05026R1

Dear Dr. Rice,

We’re pleased to inform you that your manuscript has been judged scientifically suitable for publication and will be formally accepted for publication once it meets all outstanding technical requirements.

Kind regards,

Andrea L. Conroy, PhD

Academic Editor

PLOS ONE
---

## [Editor Report · Acceptance letter]

23 Aug 2023

PONE-D-23-05026R1 

Mortality after emergency unit fluid bolus in febrile Ugandan children 

Dear Dr. Rice:

I'm pleased to inform you that your manuscript has been deemed suitable for publication in PLOS ONE. Congratulations! Your manuscript is now with our production department. 

Kind regards, 

on behalf of

Dr. Andrea L. Conroy 

Academic Editor

PLOS ONE